# The Kidney–Gut–Muscle Axis in End-Stage Renal Disease is Similarly Represented in Older Adults

**DOI:** 10.3390/nu12010106

**Published:** 2019-12-30

**Authors:** Michael S. Lustgarten

**Affiliations:** Nutrition, Exercise Physiology, and Sarcopenia Laboratory, Jean Mayer USDA Human Nutrition Research Center (HNRCA), Tufts University, Boston, MA 02111, USA; Michael.Lustgarten@Tufts.edu; Tel.: +1-617-556-3019

**Keywords:** ESRD, gut microbiome-derived uremic metabolites, aging, renal function, sarcopenia, muscle composition

## Abstract

Decreased renal function, elevated circulating levels of urea, intestinal levels of urea-degrading bacteria, and gut-derived uremic metabolites are present in end-stage renal disease (ESRD), a cohort that has reduced muscle mass and physical function, and poor muscle composition. This phenotype, defined as the kidney–gut–muscle axis, is similarly represented in older adults that do not have ESRD. The purpose of this short communication is to illuminate these findings, and to propose a strategy that can positively impact the kidney–gut–muscle axis. For example, dietary fiber is fermented by intestinal bacteria, thereby producing the short-chain fatty acids (SCFAs) acetate, propionate, and butyrate, which affect each component of the kidney–gut–muscle axis. Accordingly, a high-fiber diet may be an important approach for improving the kidney–gut–muscle axis in ESRD and in older adults that do not have ESRD.

## 1. Introduction: The Kidney–Gut–Muscle Axis in ESRD

### 1.1. Gut-Derived Uremic Metabolites Negatively Affect Skeletal Muscle

The kidney and gut microbiome may be involved in mechanisms related to the maintenance of skeletal muscle mass, composition, and physical function (kidney–gut–muscle axis). In support of this, muscle mass is reduced, muscle composition is altered (i.e., an increased deposition of lipids or adipocytes within and/or between muscle), and poor physical function is prevalent in adults younger than 65 y that have end-stage renal disease (defined as an estimated glomerular filtration rate (eGFR) less than 15 mL/min/1.73 m^2^) [1,2,3,4]. Investigating further, ESRD patients have elevated circulating levels of urea and uremic metabolites [5,6,7] in conjunction with alterations in gut microbiome composition and functions, including increased intestinal levels of urea-degrading bacteria (e.g., urease-expressing *Enterobacteriaceae*) [8,9,10]. *Enterobacteriaceae* may have an important role that connects reduced renal function with alterations in muscle mass, composition, and physical function. First, intestinal epithelial cell permeability is increased when levels of urea and urease are elevated [11,12,13,14]. Second, *Enterobacteriaceae* contain genes involved in the production of indoxyl sulfate (IS) and p-cresyl sulfate (PCS) [8], metabolites that are exclusively produced by the gut microbiome [15] that are increased in ESRD (i.e., gut-derived uremic metabolites; GDUMs) [5,7]. A more permeable gut barrier may allow for the increased passage of IS and PCS from the intestinal lumen into the blood, where they negatively affect skeletal muscle. IS and PCS accumulate within skeletal muscle [16,17], where they induce gene expression of the muscle atrophy markers myostatin and atrogin-1, thereby resulting in decreased muscle mass [16,18], or increase intramuscular lipid content, thereby promoting muscle cell insulin resistance [19]. Furthermore, in vitro exposure of C2C12 muscle cells to IS results in decreased mitochondrial function, increased oxidative stress and inflammatory signaling, and myotube atrophy [16,18,20]. In addition, treadmill endurance exercise capacity is reduced in association with elevated circulating levels of IS in a mouse model of ESRD, and supplementation with an oral adsorbent reduced IS in conjunction with improved physical function [21]. Collectively, these data suggest roles for decreased renal and intestinal barrier function, and increased circulating levels of urea, intestinal levels of urea-degrading bacteria, and gut microbiome-derived uremic metabolites on muscle mass, composition, and physical function in ESRD.

### 1.2. Short-Chain Fatty Acids Positively Affect the Kidney–Gut–Muscle Axis, but Are Reduced in ESRD

Alternatively, fecal levels of short-chain fatty acids (SCFAs), including acetate, propionate, and butyrate are reduced in a mouse model of ESRD and in human ESRD patients [22,23], an important finding because SCFAs positively affect each component of the kidney–gut–muscle axis. Propionate and butyrate are exclusively gut microbiome-derived metabolites, as they are not found in germ-free mice [24], whereas acetate is produced by humans and bacteria. Renal function is improved in rats injected with butyrate [25], butyrate impairs the progression of kidney disease [26], and acetate, propionate, and butyrate each improved renal function in an experimental model of acute kidney injury [27]. Moreover, circulating levels of urea are reduced in animal models of ESRD in response to propionate or butyrate [25,27]. However, it is important to note that there may be a threshold effect for SCFAs on renal function. Oral administration of 200 mM acetate induces renal disease, including renal swelling and inflammation, whereas these effects are not present at closer-to-physiologic doses (100 mM) [28].

Acetate, propionate, and butyrate positively affect the gut, as evidenced by decreased *Enterobacteriaceae* growth and improved intestinal barrier function following exposure to physiological levels of these SCFAs [29,30]. Although the role of SCFAs on muscle in ESRD has yet to be examined, they positively impact muscle in young animals. Muscle mass and strength were increased in germ-free mice that were fed a combination of acetate, propionate, and butyrate [31], acetate and propionate each improved treadmill endurance capacity in conventionally-raised mice [32,33], acetate reduced intramuscular triglyceride content in rabbits [34], and muscle mass was increased in pigs in response to dietary butyrate supplementation [35].

Fecal levels of acetate, propionate, and butyrate are increased following dietary fiber fermentation by gut bacteria [36], and accordingly, fiber-based interventions would be expected to positively affect the kidney–gut–muscle axis in animal models of ESRD. In support of this, fecal SCFAs are increased, renal and intestinal barrier function are improved, and circulating levels of urea and PCS are decreased in response to dietary fiber supplementation [23,37]. Although Van Hung et al. and Yang et al. [23,37] did not evaluate the effect of dietary fiber on intestinal *Enterobacteriaceae* abundance or on skeletal muscle, alternatively, intramuscular lipid content was reduced in conjunction with decreased circulating levels of PCS in response to dietary fiber supplementation with arabino-xylo-oligosaccharides (AXOS) [19]. SCFAs were not quantified in Koppe et al. [19], but fecal levels of butyrate, and the sum of acetate, propionate, and butyrate are increased in response to AXOS feeding [38,39]. Moreover, in humans diagnosed with ESRD, dietary fiber intake is associated with improved renal function and reduced circulating levels of IS and PCS [40,41,42,43]. Separately, in young, healthy mice that did not have ESRD, fecal SCFAs were increased in response to a high-fiber diet in conjunction with increased muscle mass and treadmill endurance capacity, when compared with low-fiber fed mice [32]. Collectively, these studies suggest that dietary fiber supplementation may be an important approach for increasing fecal SCFAs, thereby positively impacting the kidney–gut–muscle axis in ESRD.

## 2. The Kidney–Gut–Muscle Axis in Older Adults

### 2.1. Decreased Renal Function, Increased Circulating Levels of Urea, Intestinal Urea-Degrading Bacteria, and GDUMs Are Also Present in Older Adults that Do Not Have ESRD

Similarly, a kidney–gut–muscle axis may be present in older adults that do not have ESRD. Reductions in muscle mass and physical function, and poor muscle composition are prevalent in older adults [44,45], and kidney function decreases during aging from average eGFR values of 120 mL/min/1.73 m^2^ in 20-year-olds to less than 60 mL/min/1.73 m^2^ in adults older than 70 y [46]. Correspondingly, circulating levels of urea (i.e., blood urea nitrogen) increase during aging [46], which may provide the stimulus for an increase in urea-degrading intestinal bacteria, while impairing intestinal barrier function. In support of this, *Enterobacteriaceae* are higher in subjects older than 60 y, when compared with younger subjects (19–45 y) [47,48,49], an important finding because *Enterobacteriaceae* are associated with frailty, a reduced percentage of whole body lean mass, and worse physical function in older adults [50,51,52]. In addition, colonic permeability increases during aging in mice and monkeys [53,54], but studies in older adult humans have yet to be published. As mentioned in Section 1.1, *Enterobacteriaceae* contain genes involved in the production of IS and PCS [8], and accordingly, serum levels of these metabolites are higher in adults older than 65 y, when compared with younger subjects [55,56], and increase during aging [57], evidence that identifies IS and PCS as age-related, gut-derived uremic metabolites. When considering that IS and PCS negatively affect muscle mass and composition in cells and young mice, these data suggest a negative role for IS and PCS on muscle in older adults that do not have ESRD. It is important to note that studies aimed at reducing circulating levels of IS, PCS, or other GDUMs and evaluating the resultant effect on muscle in aged animals or humans have yet to be reported.

### 2.2. Age-Related GDUMs Other than Indoxyl Sulfate and p-Cresol Sulfate May Negatively Affect Muscle

*Enterobacteriaceae* also contain tyrosine phenol-lyase, which converts the amino acid tyrosine into pyruvate, ammonia, and phenol [58,59]. Phenol is sulfated in colonocytes (e.g., phenol sulfate; PS), absorbed into the bloodstream, and normally excreted into the urine [60], but it systemically accumulates when renal function declines [61]. PS is found in conventionally-raised, but not germ-free mice [15], and its levels increase during aging [57,62], findings that collectively identify it as an age-related, gut microbiome-derived uremic metabolite. Similarly, phenylacetylglutamine is produced via the colonic bacterial degradation of phenylalanine, followed by conjugation with glutamine in the gut mucosa to form phenylacetylglutamine (PAG) [63,64]. Circulating levels of PAG increase in conjunction with decreased renal function [65,66], and during aging [57,62], findings that similarly identify it as an age-related GDUM. Higher circulating levels of PS and PAG were associated with worse muscle composition in older adults that did not have ESRD (average eGFR, 73.1 mL/min/1.73 m^2^) [67]. Additionally, cinnamoylglycine (CG) is a GDUM: its levels increase when renal function is reduced [68], and it is found in conventionally-raised, but not germ-free mice [15]. Although age-related changes for cinnamoylglycine have yet to be published, it is associated with poor muscle quality in both young and older subjects that did not have ESRD [69,70]. When considering these findings, studies aimed at evaluating a causative role for the age-related, gut microbiome-derived uremic metabolites, PS, PAG, and CG on muscle mass, composition, and physical function are of interest.

### 2.3. Fecal SCFAs Are Reduced in Older Adults

Whereas the gut-derived uremic metabolites, IS, PCS, PS, and PAG increase during aging, in contrast, fecal levels of SCFAs are decreased in older adults, when compared with middle-aged and young subjects [49,71,72]. As mentioned earlier, SCFAs positively affect renal and intestinal barrier function, are associated with reduced circulating levels of urea and gut microbiome-derived uremic metabolites, decrease *Enterobacteriaceae* growth, and improve muscle mass, composition, and physical function in young animals. Accordingly, SCFAs would be expected to positively affect the kidney–gut–muscle axis in aged animals or in older adult humans. To date, only one study has partially explored this hypothesis: dietary supplementation with butyrate increased the percentage of whole body lean mass and prevented intramuscular fat accumulation in aged mice [73], but renal and intestinal barrier function, circulating levels of urea and GDUMs, and *Enterobacteriaceae* abundance were not evaluated.

Based on the data presented in this review, a high-fiber diet would be expected to increase fecal SCFAs, which would then be involved in mechanisms that positively affect the kidney–gut–muscle axis, but this hypothesis has yet to be comprehensively studied. For example, in older adult humans, handgrip strength is improved in response to dietary supplementation with fiber [74], and fiber intake is positively associated with physical functioning [75,76], but alterations in gut microbiome composition, fecal SCFAs, renal and intestinal barrier function, and circulating levels of urea and GDUMs were not investigated. Accordingly, studies aimed at the combined investigation of a high-fiber diet on each of the components of the kidney–gut–muscle axis in older adults are of interest.

## 3. Concluding Remarks

A kidney–gut–muscle axis, including decreased renal function, increased circulating levels of urea, intestinal levels of *Enterobacteriaceae*, and gut-derived uremic metabolites, and reduced muscle mass, physical function, and poor muscle composition is present in ESRD, a phenotype that is similarly represented in older adults that do not have ESRD. Although a few studies have investigated components of the kidney–gut–muscle axis in animal models of ESRD, interventions aimed at comprehensively evaluating this hypothesis in older adults have yet to be published. Based on evidence in animal models and human ESRD patients, dietary fiber supplementation may be an important approach for increasing fecal SCFAs, which may positively impact the kidney–gut–muscle axis in older adults. Accordingly, future studies aimed at exploring this hypothesis are of interest.

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
