# Peer review of "The Kidney–Gut–Muscle Axis in End-Stage Renal Disease is Similarly Represented in Older Adults"

_nutrients, 2019, doi:10.3390/nu12010106_

Round 1

Reviewer 1 Report

The paper is well written.

1 relatively minor points only

Renal function is known to decline with age; this may not even be reflected by serum creatinine in mild cases. Authors wish to use a broader language, potentially implying that some of the changes - observed in older animals/humans and similar to ESRD - may be generated by common theme of reduced renal function. The Authors may use the term Chronic Kidney Disease (CKD), when referring to modestly reduced renal function in human To section 2.1 and 2.2.: It should be worth pointing that peritoneal dialysis (PD) provides much less small solute clearance than hemodialysis (HD) - yet, PD achieves good uremic control, as well. As it has been pointed out before [Med Hypotheses. 2017 Oct;108:128-132. doi: 10.1016/j.mehy.2017.09.005.; PMID: 29055386], this probably can be attributable to the fact that PD preferentially providers clearance to the abdominal cavity, i.e., the gastrointestinal compartment, known to generate most of the uremic toxins.

Author Response

1. Renal function is known to decline with age; this may not even be reflected by serum creatinine in mild cases. Authors wish to use a broader language, potentially implying that some of the changes - observed in older animals/humans and similar to ESRD - may be generated by common theme of reduced renal function. The Authors may use the term Chronic Kidney Disease (CKD), when referring to modestly reduced renal function in human To section 2.1 and 2.2.

-As you mentioned, and as indicated in the manuscript, renal function declines during aging from eGFR values of 120 mL/min/1.73 m2 in 20 year olds to less than 60 in adults older than 70y (Wang et al. 2017, doi: 10.1016/j.jbi.2017.11.003). A diagnosis of CKD is made when eGFR is less than 60 mL/min/1.73 m2 for more than 3 months (Kidney Disease: Improving Global Outcomes (KDIGO) CKD Work Group, doi: 10.1038/kisup.2012.73). However, the amount of time that the older adult group of Wang et al. had eGFR values < 60 was not reported, and accordingly, a diagnosis of CKD may not be applicable. To account for this finding, for the older adult data in the manuscript (Sections 2.1 – 2.3), I referred to the presence of reduced renal function, rather than a diagnosis of CKD.

2. It should be worth pointing that peritoneal dialysis (PD) provides much less small solute clearance than hemodialysis (HD) - yet, PD achieves good uremic control, as well. As it has been pointed out before [Med Hypotheses. 2017 Oct;108:128-132. doi: 10.1016/j.mehy.2017.09.005.; PMID: 29055386], this probably can be attributable to the fact that PD preferentially providers clearance to the abdominal cavity, i.e., the gastrointestinal compartment, known to generate most of the uremic toxins.

-The idea that PD may positively impact the kidney-gut-muscle axis is indeed interesting, especially when considering the negative effects of the gut-derived uremic metabolites, indoxyl sulfate and p-cresol sulfate, on muscle. However, note that PD patients have higher intestinal levels of Enterobacteriaceae when compared with healthy controls (Crespo-Salgado et al. 2016, doi: 10.1186/s40168-016-0195-9). Based on the evidence presented in the manuscript, interventions aimed at reducing, not increasing Enterobacteriaceae would be expected to positively affect the kidney-gut-muscle axis. Accordingly, I did not include use of PD in the revised manuscript.

Reviewer 2 Report

Dear Author, 

Firstly, I would like to congratulate you on a brilliant paper. I feel fellow researchers in this field would benefit from a mention of the following areas in the context of the kidney-gut-muscle axis:

Microbial dysbiosis seems to play a role in ESRD. Would you be able to provide a brief account of this relationship and how this might contribute to the modulation of the kidney-gut-muscle axis. Increased intestinal permeability or "leaky gut" has also been seen in ESRD (e.g. see Lau and Vaziri: 10.1053/j.jrn.2017.02.010). A succinct critical appraisal of these findings as an expert in the ESRD field would be welcome by the intended audience.

Thank you in advance for considering my feedback on your excellent work.

With kind regards,

MDPI Reviewer 

Author Response

Firstly, I would like to congratulate you on a brilliant paper.

-Thanks!

1. I feel fellow researchers in this field would benefit from a mention of the following areas in the context of the kidney-gut-muscle axis: Microbial dysbiosis seems to play a role in ESRD. Would you be able to provide a brief account of this relationship and how this might contribute to the modulation of the kidney-gut-muscle axis.

-As you mentioned, microbial dysbiosis has been reported in ESRD, and additionally occurs in older adults. For example, as reported in the manuscript, Enterobacteriaceae are increased both ESRD patients and in older adults that do not have ESRD. However, in contrast with reporting global bacterial population and/or functional differences, which may or may not be involved a given mechanism, in the manuscript I focused on gut-derived metabolic products, including GDUMs (i.e. IS, PCS, PS, PAG, and CG) and SCFAs, which may have a more direct role on the kidney-gut-muscle axis.

2. Increased intestinal permeability or "leaky gut" has also been seen in ESRD (e.g. see Lau and Vaziri: 10.1053/j.jrn.2017.02.010). A succinct critical appraisal of these findings as an expert in the ESRD field would be welcome by the intended audience.

-Data on the role of intestinal permeability in ESRD and in older adults that do not have ESRD has been added to the revised manuscript.

Reviewer 3 Report

As the topic described by Dr. Lustgarten “The Kidney-Gut-Muscle Axis In End-Stage Renal Disease Is Similarly Represented In Older Adults” is interesting and promising. There are some suggestions need to be modified carefully.

(1) In this short communication paper, it is very hard to find the objectives and questions being address. As a short communication it might have focused on particular aspect of a problem or a new finding. However, in abstract section author has stated it as mini-review.

(2) Introduction and background are need to be elaborate.

(3) There is lack of molecular approach to justify the author conclusion, because SCFAs acts as double sided swords. SCFAs, when administered with higher then physiological dose it may cause dysregulation of T cell responses and tissue inflammation in the renal system.

(3) Very few study and molecular and immunological justification to represent end-stage renal disease (ESRD) phenotype as older adults that do not have ESRD.

(4) English is not good enough and needs to be improvised. At several sentences both abbreviation, IS, and full form indoxyl sulfate is used as well. The structure of manuscript need to be improvised.

Author Response

(1) In this short communication paper, it is very hard to find the objectives and questions being address. As a short communication it might have focused on particular aspect of a problem or a new finding. However, in abstract section author has stated it as mini-review.

-The manuscript was intended to be submitted as a short communication paper, not a mini-review. In the revised submission, I removed “mini-review” from the Abstract, and apologize for the confusion.

In terms of the objective and questions being addressed, the purpose of the manuscript is to illuminate the similarities for the kidney-gut-muscle axis in ESRD patients with older adults that do not have ESRD, as this is a novel hypothesis that has not yet been suggested in the published literature.

(2) Introduction and background are need to be elaborate.

-In this short communication paper, the introduction and background are used to define the components of the kidney-gut-muscle axis in ESRD patients, including the role of gut-derived uremic metabolites (Section 1.1) and SCFAs (Section 1.2). The similar phenotype for the kidney-gut-muscle axis in older adults that do not have ESRD is then reported in Section 2.

(3) There is lack of molecular approach to justify the author conclusion, because SCFAs acts as double sided swords. SCFAs, when administered with higher then physiological dose it may cause dysregulation of T cell responses and tissue inflammation in the renal system.

-Thank you for illustrating this important finding. Accordingly, to Section 1.2 of the revised manuscript I added, “However, it is important to note that there may be a threshold effect for SCFAs on renal function. Oral administration of 200 mM acetate induces renal disease, including renal swelling and inflammation, whereas these effects are not present at smaller doses (100 mM).”

(4) Very few study and molecular and immunological justification to represent end-stage renal disease (ESRD) phenotype as older adults that do not have ESRD.

-Although it is beyond the scope of this short communication paper, circulating levels of several acute-phase and inflammatory markers, including IL-6, C-reactive protein, and fibrinogen increase as renal function progresses towards ESRD, and additionally, levels of these analytes increase during aging (Ferrucci et al., PMID 15572589; Gupta et al. 2012, PMID 23024164). The focus of the manuscript is not to report all of the possible evidence that connects the ESRD phenotype with that found during aging, but to highlight the overlapping data for the kidney-gut muscle axis in ESRD and in older adults that have age-related reductions in kidney function.

(5) English is not good enough and needs to be improvised. At several sentences both abbreviation, IS, and full form indoxyl sulfate is used as well. The structure of manuscript need to be improvised.

-In the revised manuscript, abbreviations for indoxyl sulfate (IS) and p-cresol sulfate (PCS) were used ubiquitously throughout the text.